# Static Globularization Behavior and Artificial Neural Network Modeling during Post-Annealing of Wedge-Shaped Hot-Rolled Ti-55511 Alloy

**DOI:** 10.3390/ma16031031

**Published:** 2023-01-23

**Authors:** Liguo Xu, Shuangxi Shi, Bin Kong, Deng Luo, Xiaoyong Zhang, Kechao Zhou

**Affiliations:** 1State Key Laboratory of Powder Metallurgy, Central South University, Changsha 410083, China; 2Hunan Xuangtou Goldsky Titanium Metal Co., Ltd., Changsha 410221, China; 3Xiangtan Iron & Steel Group Co., Ltd., Xiangtan 411104, China

**Keywords:** Ti-55511 alloy, wedge-shaped hot-rolling, annealing, static globularization, artificial neural network (ANN) –

## Abstract

The globularization of the lamellar α phase by thermomechanical processing and subsequent annealing contributes to achieving the well-balanced strength and plasticity of titanium alloys. A high-throughput experimental method, wedge-shaped hot-rolling, was designed to obtain samples with gradient true strain distribution of 0~1.10. The samples with gradient strain distribution were annealed to obtain the gradient distribution of globularized α phase, which could rapidly assess the globularization fraction of α phase under different conditions. The static globularization behavior under various parameters was systematically studied. The applied prestrain provided the necessary driving force for static globularization during annealing. The substructure evolution and the boundary splitting occurred mainly at the early stage of annealing. The termination migration and the Ostwald ripening were dominant in the prolonged annealing. A backpropagation artificial neural network (BP-ANN) model for static globularization was developed, which coupled the factors of prestrain, annealing temperature, and annealing time. The average absolute relative errors (AARE) for the training and validation set are 3.17% and 3.22%, respectively. Further sensitivity analysis of the factors shows that the order of relative importance for static globularization is annealing temperature, prestrain and annealing time. The developed BP-ANN can precisely predict the static globularization kinetic curves without overfitting.

## 1. Introduction

Near-β titanium alloys have been widely applied in the aviation industry due to their characteristics of good processability, strong mechanical properties, and light weight [1,2]. The mechanical properties of near-β titanium alloys are significantly influenced by the morphology, size, and distribution of the α phase [3,4]. Lamellar α, as a typical microstructure in titanium alloys, has good performance in fracture toughness, but its performance in plasticity is not ideal [5,6]. The equiaxed α, which can be obtained by breaking down the lamellar α, is desirable due to its good ability to match the strength and plasticity [7]. However, the lamellar α cannot be globularized directly by heat treatment due to the thermal stability of the semicoherent phase interface between α and β phase. Thermal processing and post-annealing in the α+β phase region are usually conducted to achieve the globularization of lamellar α phase [8]. In addition, the static globularization during post-annealing can further improve the globularization fraction and obtain more uniform microstructure [9].

Static globularization during annealing is a complex process, which includes the evolution of substructures, phase transformation, and the coarsening of the α phase. Boundary splitting, termination migration, and Ostwald ripening have been applied to explain the static globularization behavior during annealing [10]. Boundary splitting is a process of α/α sub-boundaries transforming into the α/β phase interface at the driving of surface energy [11]. At prolonged annealing, diffusion-controlled termination migration and Ostwald ripening are two main mechanisms that lead to the coarsening of the α phase together. Moreover, the dissolution of the termination tip also reduces the length-to-diameter ratio of lamellar α [12]. There is an interaction between boundary splitting and termination migration. For example, Fan et al. [13] proposed that boundary splitting and microstructure coarsening were competitive relationships during prolonged annealing.

The static globularization kinetic model can visually reflect the relationship between microstructure characteristics and process parameters, which is conducive to the efficient regulation of globularized α phase. Zherebtsov et al. [14] developed two different physical models to predict the static globularization time, which were based on boundary splitting and termination migration. Fan et al. [13] fitted the kinetic curves of TA15 titanium alloy during annealing using the JMAK equation and the asymptotic equation. Xu et al. [15,16] applied the JMAK equation, the modified JMAK equation, and the ANN model to simulate the static globularization behavior of TC17 alloy, respectively. The results showed that the ANN model had more predicted accuracy than JMAK equation. The ANN models have excellent ability to deal with complex relationships and have good prospects in material science fields, such as alloy design and phase transformation [17,18,19], microstructure prediction [20,21], and constitutive models [22]. Seyed Salehi et al. [23] developed an ANN model to predict the static recrystallization volume fraction of AA5083 after hot rolling, and the predicted kinetic curves had a good consistency with the experimental results. Liu et al. [24] established a BP-ANN model to analyze the factor sensitivity, aiming to optimize the microstructure.

In previous work, many researchers obtained microstructure characteristics from individual samples. Comparatively, high-throughput experimental methods allow researchers to quickly obtain microstructure characteristics for modeling purposes. The thermal processing of wedge-shaped sheets can rapidly obtain samples with different deformation strains [25,26]. In this work, a high-throughput experimental method combined with ANN modeling was conducted to study the static globularization behavior of hot-rolled Ti-55511 alloy. The fraction of the globularized α at different processing parameters was rapidly obtained from the high-throughput samples. Furthermore, the evolution of sub-boundaries was described with the support of the EBSD technique. Based on quantitative data, an ANN model was developed. The sensitivity level was introduced to value the importance of process parameters. The prediction of kinetic curves will guide the controlling of the annealed microstructure of Ti-55511 alloy.

## 2. Experimental Section

### 2.1. Materials

The received material was TC18 forged bar, which was provided by Xiangtou Goldsky Titanium Industry (Changde, China). Its nominal composition (wt.%) was 5.16 Al, 4.92 Mo, 4.96 V, 1.10 Cr, 0.98 Fe, and balanced Ti. The β transit temperature was about 875 °C. The schematic illustration of this experiment is shown in Figure 1. A rectangular sample of 60 mm × 30 mm × 11.2 mm was wire-cut from the forged bar. To obtain a lamellar microstructure, the sample was held at 905 °C for 30 min in a vacuum tube furnace, followed by cooling to 350 °C at 3 °C/min, and finally air-cooled.

As Figure 2 shows, the interweaved α lamellae have an average thickness of 0.32 μm. The Burgers orientation relationship (BOR) between α and β phase is restricted by (0001)_α_//(110)_β_; (11–20)_α_//(111)_β_ [27]. The wedge-shaped sheets, with continuous thickness variation from 3 to 8 mm, width of 30 mm, and length of 60 mm, were cut from the rectangular sample.

### 2.2. Wedge-Shaped Hot Rolling and Post-Annealing

The wedge-shaped sheets were hot-rolled at 750 °C for three passes. The samples were heated at 750 °C for 30 min before hot rolling and at 750 °C for 5 min between each pass. The corresponding maximum thicknesses of the rolled sheets after each pass were 5.6, 4.2, and 3.2 mm, respectively. After the final pass, the sheets were air-cooled. The thickness variation of wedge-shaped sheets was translated into gradient strain distribution along the roll direction (RD), which was an effective method to prepare high-throughput samples [28]. To precisely describe the true strain distribution of the hot-rolled sheets, the hot-rolling process was simulated by finite element method (FEM) with the support of DEFORM-3D software. Long strip samples with gradient strain distribution were wire-cut from hot-rolled sheets, and then were annealed at 750, 775, 800, and 825 °C for 10, 30, 60, 120, 240, and 480 min, respectively.

### 2.3. Microstructure Quantitative Analysis

The observation samples, corresponding to true strains of 0.32, 0.57, 0.83, and 1.06, were wire-cut from different areas on the annealed samples. The observation areas of SEM and EBSD were RD-TD plane (intermediate thickness). The SEM specimens were ground, polished, and etched, which were investigated in backscatter mode on a Tescan Mira4 scanning electron microscope. The EBSD specimens were electropolished in a solution of 10% perchloric acid, 30% n-butanol, and 60% methanol at about −30 °C. EBSD characterization was performed on a Helios Nano Lab G3 UC scanning electron microscope (Changsha, China).

In the EBSD graphs, the black lines represent high-angle grain boundaries (HAGBs), where the orientation difference is over 15°. The gray lines represent low-angle grain boundaries (LAGBs), where the orientation difference is between 2° and 15°. The SEM micrographs were quantitatively analyzed by Image pro plus 6.0 software to obtain the globularization fraction of α phase after hot rolling and annealing. The α phase with aspect ratio less than 2.5 is defined as globularized α. The globularization fraction can be expressed by Equation (1):(1)$$f_{\alpha }=F_{g}/F_{\alpha }$$ where $F_{g}$ is the area proportion occupied by globularized α (μm^2^) in a specific observation field, and
$F_{\alpha }$ is the area proportion of α phase (μm^2^) in the same observation field. Each specimen was quantitatively analyzed from four different SEM graphs at least, and the average value was taken as the result.

### 2.4. ANN Modeling and Prediction

A double-hidden-layers BP-ANN model with 4 hidden neurons was used to model the static globularization behavior. The globularization fraction was defined as a function of prestrain, annealing temperature, and annealing time. The process of finding the optimum solution is based on function signal forward feed and error signal backpropagation, as shown in Figure 3.

Four statistical indicators are employed to quantitatively assess the performance of the model, which included the correlation coefficient ^®^, the relative error (RE), the average absolute relative error and the standard deviation (S.D.):(2)$$R=\overset{n}{\underset{i=1}{\sum }}\left(f_{\alpha e}-{\bar{f}}_{\alpha e}\right)\left(f_{\alpha p}-{\bar{f}}_{\alpha p}\right)/\sqrt{\overset{n}{\underset{i=1}{\sum }}{\left(f_{\alpha e}-{\bar{f}}_{\alpha e}\right)}^{2}{\left(f_{\alpha p}-{\bar{f}}_{\alpha p}\right)}^{2}}$$
(3)$$RE=100\%\times \left(f_{\alpha p}-f_{\alpha e}\right)/f_{\alpha e}$$
(4)$$AARE=\left[\overset{n}{\underset{i=1}{\sum }}|100\%\times \left(f_{\alpha p}-f_{\alpha e}\right)/f_{ae}|\right]/n$$
(5)$$S.D.=\sqrt{[\overset{n}{\underset{i=1}{\sum }}{\left(RE_{i}-\bar{RE}\right)}^{2}]/\left(n-1\right)}$$ where *n* is the number of total samples, $f_{\alpha e}$ is the experimental globularization fraction, and $f_{\alpha p}$ is the globularization fraction predicted by the ANN model. ${\bar{f}}_{\alpha e}$, ${\bar{f}}_{\alpha p}$, and $\bar{RE}$ are the average values of $f_{\alpha e}$, $f_{\alpha p}$, and RE, respectively.

## 3. Results and Discussion

### 3.1. Hot-Rolled Microstructure

Figure 4a shows the hot-rolled sheet and its strain distribution in the mid-thickness plane. The FEM results show that the sheets have a gradient strain distribution in the RD, and a uniform strain distribution in the transverse direction (TD). The lengths of the experimental and simulated rolled sheet are 97.13 and 96.75 mm, respectively, and the relative error at size is 0.39%. Thus, the true strain distribution on the hot-rolled sheet can be described by FEM precisely. As shown in Figure 4b, the range of true strain field in hot-rolled sheets is 0.00~1.10. The distances from the four observation areas to the left are 15, 30, 55, and 85 mm, respectively, corresponding to the true strain of 0.32, 0.57, 0.83, and 1.06, respectively.

Figure 5 demonstrates the inverse pole figure (IPF) color maps, the phase maps, and the corresponding pole figures of two phases at a strain of 0.32 and 0.83. When the strain is 0.32, most of the α phase still keeps a lamellar morphology. Some orientation differences within lamellar α are produced, corresponding to the color change in the IPF color map, marked by gray arrows (Figure 5a). When the strain increases to 0.83, the lamellar α is bent, and large orientation differences are generated within lamellar α, marked by gray arrows (Figure 5b). Meanwhile, some sub-boundaries are formed, which are marked by black arrows. Due to the high deformation rate of hot rolling, a large number of substructures remain, which highlight the microstructure evolution during annealing. Different numbers of remaining substructures provide different driving forces for subsequent static globularization.

As illustrated in Figure 5e, the clusters of the α phase in the (0001)_α_ and (11–20)_α_ pole figure have a counterpart in (110)_β_ and (111)_β_, respectively. At light deformation, the α and β phases can rotate synergistically with keeping basic BOR. When the strain increases, the coordinated rotation between the α and β phase becomes difficult, and the correspondence of BOR is obviously deviated, as shown in Figure 5f. Wang et al. [29] reported the pole figures of the α and β phase under different strains in TA15 alloys, and the orientation relationship deviation of the two phases became larger when the strain increased. Meanwhile, the scattered (0001)_α_ pole figure of α phase proves that more substructures are generated at a higher strain (Figure 5f). The destruction of the BOR and the generation of new sub-boundaries during hot deformation within the α phase will boost the penetration of the β phase during the subsequent annealing [30].

### 3.2. Annealed Microstructure

#### 3.2.1. Effect of Prestrain

Figure 6a–d illustrates the microstructure annealed at 775 °C for 30 min with different prestrain. After annealing for 30 min, the energy in the deformed microstructure is released, which promotes the static globularization of the α phase. As the quantitive result shows (Figure 6e), the globularization fractions of the α phase are 11.8%, 23.5%, 31.3%, and 37.7% when the prestrains are 0.32, 0.57, 0.83, and 1.06, respectively. The globularization fraction in the annealed microstructure improves with increasing pre-strain. When the prestrain is 0.32 (Figure 6a), the annealed microstructure consists of residual lamellar α and few globularized α. The thermal groovings on lamellar α are not enough, leading to a slight improvement in the globularization fraction (from 6.5% to 11.4%). When the prestrain increases, the large sub-boundaries in the initial lamellar α can serve as the locations for boundary splitting [31]. The thermal groovings of lamellar α are more significant, and the fraction of short bar-like and globularized α is improved. At the prestrain of 1.06 (Figure 6d), large remaining substructures provide a larger driving force to static globularization. The deformed lamellar α separates into a large number of short bar-like α and globularized α. Compared to the hot-rolled microstructure, there is a large improvement in the globularization fraction (from 10.5% to 37.7%).

#### 3.2.2. Effect of Annealing Yime

Figure 7 shows the effect of annealing time on the microstructure of deformed Ti-55511 alloy annealed at 775 °C with a prestrain of 0.83. The coarsening and shortening of the α phase are the main changes in microstructure during the prolonging of annealing time. As the quantitative analysis result shows (Figure 7f), the main increase in the globularization fraction is concentrated in the first 60 min, which changes from 8.4% to 36% (Figure 6e,f). Subsequently, it improves slowly, with the annealing time extending from 60 min to 480 min (from 36% to 45.3%). A similar microstructure evolution had been found in TC17 alloy by Xu et al. [32] and Pang et al. [33]. Corresponding to the sharp increase in the globularization fraction, a large part of the deformed lamellar α separates into short bar-like α and globularized α after annealing at 775 °C for 60 min, as shown in Figure 7b. In the initial stage of static globularization, the boundary splitting mechanism results in the quick separation of lamellar α with the further release of deformed distortion energy [34].

Compared to the initial stage, the increase in the static globularization fraction in the middle and latter stages of annealing is relatively slower, but the coarsening of the α phase is obvious, as shown in Figure 7b–e. In this stage, the coarsening mechanisms are termination migration and Ostwald ripening, which are only conducted by the migration of atoms. The termination tip of lamellar α has higher chemical potential energy than other positions of the flat interface. The solution atoms migrate from the termination tip to adjacent flat surfaces, aiming to reduce the energy of the system [35]. In this study, the termination migration and the Ostwald ripening will lead to three changes in microstructure: 1—the boundary splitting being suppressed [13]; 2—the coarsening of lamellar α; 3—the dissolution of the termination tip and the decrease in the aspect ratio of α plates; 4—the decrease in the amount of α phase. Under the action of coarsening mechanisms, a microstructure with more uniform globularized α was obtained after annealing at 775 °C for 480 min, as shown in Figure 7e. Unlike the boundary splitting mechanism, the coarsening mechanism is the relatively slower migration process of atoms, which leads to slight changes in the globularization fraction. Furthermore, the termination migration mechanism occurs all the time during the static globularization process. As shown in Figure 7a, due to the shortened distance of termination migration, the short bar-like α and globularized α are thicker than the residual lamellar α.

It is noteworthy that even after a long annealing time, some thermal groovings could not completely penetrate the lamellar α. Under the combined action of boundary splitting, termination migration and Ostwald ripening, the chain-like α phase is formed, as shown in Figure 7d,f. This phenomenon seems to support the opinion proposed by Fan et al. [13] in their study of TA15 alloy. The boundary splitting and microstructure coarsening are two competing mechanisms during static globularization. Li et al. [36] found a similar phenomenon in TC17 alloy. The relatively slow migration of atoms in the β phase led to the limitation of grooving on lamellar α, and chain-like α formed during prolonged annealing. Furthermore, the substructure rotation during annealing may cause the formation of chain-like α, according to the research of Weiss et al. [37]. The suppression of boundary splitting also is the reason why full globularization cannot be achieved when annealing at relatively low annealing temperatures for a certain time.

#### 3.2.3. Effect of Annealing Temperature

Figure 8a–d shows the microstructure of Ti-55511 alloy deformed to a strain of 0.83 and annealed at different temperatures for 30 min. With the increase in annealing temperature, the α→β phase transformation is accelerated, and it causes a decrease in the α phase fraction (from 27% to 9.8%), as shown in Figure 8e. When the annealing temperature increases from 750 °C to 825 °C, the globularization fraction of the α phase is improved from 21.7% to 54.5%. At lower annealing temperatures, such as 750 °C, lots of thermal groovings can be observed on the lamellar α, which illustrates that most of the α phase is in the static globularization stage of boundary splitting.

Comparatively, after annealing at 825 °C for 30 min, most of the lamellar α is separated. Thus, the static globularization of the α phase is boosted by increasing temperature. The obvious improvement in the globularization fraction is attributed to the quicker separation of deformed lamellar α at the higher annealing temperature. High annealing temperature boosts the migration rate of elements, which is beneficial to boundary splitting. Moreover, the process of termination migration is also promoted when the annealing temperature increases.

### 3.3. Evolution Mechanism of Sub-Boundaries

Figure 9a,b shows the IPF color map and phase map of the microstructure deformed to a strain of 0.83 and annealed at 775 °C for 30 min. As shown in the phase map, more globularized α and short bar-like α are observed in annealed microstructure, which is the result of lamellar α separating along the sub-boundaries. Figure 9c shows the variation of the misorientation angle along the arrow L_1_ in Figure 9a. The misorientation angles between sub-boundaries are about 10°, which belong to LAGBs. Similarly, the misorientation angle along arrow L_2_ is shown in Figure 9d, and the sub-boundaries between grains belong to HAGBs. In this work, because of the instability of the α/α sub-boundaries, the LAGBs and HAGBs can serve as the location for the penetration of the β phase. The process of boundary splitting is controlled by atom migration and phase transformation. Pang et al. [38] indicated that the process of boundary splitting could be described as β stabilized elements diffusing to α/α sub-boundaries, α/α sub-boundaries transforming into β (matrix) phase, and the α phase separating along the α/α sub-boundaries. Sharma et al. [39] studied the mass transport rate for the boundary-splitting mechanisms based on Ti-47Al alloy, and the results indicated that higher heat treatment temperature was beneficial for mass transport. When the annealing temperature (775 °C) is higher than the deformation temperature (750 °C), the high element diffusion rate and phase transformation of α→β can accelerate the separation of lamellar α. Though the LAGBs have lower interface energy than HAGBs, the boundary splitting at LAGBs can be promoted by high annealing temperatures. With increasing annealing temperature, this trend seems to be more obvious, as shown previously in the quantitative analysis in Section 3.2.3 (Figure 8).

As shown in Figure 10, the fraction improvement of the average misorientation angle confirms that the deformed lamellar α will furtherly perform static recrystallization with the consumption of deformation substructures. The fraction of HAGBs of the deformed lamellar microstructure is 17.35%, and the fraction increases to 24% after annealing at 775 °C for 30 min. Chen et al. [31] studied the evolution of new α grain boundaries during hot deformation and annealing, and also proposed that post-annealing would promote the formation of new sub-boundaries, which was beneficial for boundary splitting. During annealing, the separation of lamellar α and the formation of sub-boundaries happen simultaneously. The improvement of misorientation can promote boundary splitting and lead to the further separation of lamellar α.

### 3.4. Artificial Neural Network Modeling and Prediction in Static Globularization

The BP-ANN and its variants have excellent ability to process the nonlinear relationships between input data and output data. The genetic algorithm (GA) is usually used to optimize the modeling process of BP-ANN, which is a global searching process of finding the best initial fitness value of the optimal weights and thresholds. It can help the BP-ANN to find the optimal solution and to avoid a local optimum, which is achieved by adjusting the weights and thresholds until achieving the target error or approaching the maximum steps, as shown in Figure 11.

#### 3.4.1. Modeling of GA/BP-ANN

In this work, a GA/BP-ANN was used to model the static globularization behavior of Ti-55511 during annealing. In the genetic algorithm progress, the population size, the cross-selection probability, and the fraction of mutation were set as 90, 0.9, and 0.2, respectively. In the ANN progress, the transfer functions of the input layer, hidden layer, and output layer were tansig, tansig, and purelin, respectively. The optimization algorithm was the Levenberg-Marquardt algorithm and the minimum error tolerance was set as 0.0001. The data from annealing at different temperatures for 30 min were selected as the validation set (16 sets), and the remaining data (80 sets) were selected to train the GA/BP-ANN. The specific values for training and validation are shown in Appendix A, Table A1. Before the training and validating of ANN, the data values are normalized to specific data ranges (from −1 to 1), which is achieved by Equation (6):(6)$$X_{i}=2\frac{\left(X-X_{\min}\right)}{X_{\max}-X_{\min}}-1$$where *X* is the experimental data value, $X_{\max}$ is the maximum value of *X*, $X_{\min}$ is the minimum value of *X*, and the $X_{i}$ is the corresponding value of *X* after normalization.

Figure 12 shows that the training results of ANN and the regression analysis and error analysis are applied to evaluate the precision of the ANN model. As shown in Figure 12a, there is a good linear relationship between the experimental training set and the predicted training set. In addition, the R^2^ and the AARE between the training set and predicted data are 0.99414 and 3.17%, respectively. The data points are concentrated near the best-fitting line, most of which are located within the 10% deviation line. Correspondingly, the RE values are between the range of −15.42% to 12.07%. For the validation set, the R^2^ and the AARE between experimental data and predicted data are 0.99111 and 3.22%, respectively. The relative errors are between −10.67% and 4.32%, which is shown in Figure 12b. Furthermore, the S.D. of the training set and validation set are 3.15 and 4.21, respectively, which shows that the ANN has good error stability without the phenomenon of local optimum. As demonstrated in Figure 12c,d, the values of experimental data and predicted data are almost overlapped, and the fluctuation of REs is relatively stable within a specific range. The comparison between experimental data and predicted data illustrates that the ANN has a good generalization ability to model the relationship among prestrain, annealing temperature, annealing time, and globularization fraction while keeping a high prediction accuracy.

#### 3.4.2. Application of Developed GA/BP-ANN

The sensitivity factor was introduced to quantitatively measure the significance of processing parameters. The individual input factors vary at different rates (+1%, +3%, +5%, −1%, −3%, −5%), and the corresponding outputs were calculated by the GA/BP-ANN model. The sensitivity level factors for strain, temperature, and annealing time were calculated according to Equation (7), respectively [24].
(7)$$Sensitivity\;Level\;of\;X_{i\;}\left(\%\right)=\frac{1}{n}\overset{n}{\underset{j=1}{\sum }}\left(\frac{\%\;change\;in\;output}{\%\;change\;in\;input}\right)\times 100$$
where *n* equals 96, which is the total number of data sets used in this work.

The sensitivity level of the annealing temperature is much higher than the prestrain and the annealing time (Figure 13). It is understandable that the annealing temperature is the most important factor. Static globularization is a process of distortion energy release, atom diffusion, and phase transformation, which is greatly influenced by annealing temperature [32]. Increasing the annealing temperature will significantly promote the process of static globularization [40]. It can be deduced from sensitivity levels at different rates that the effect of high temperature is greater than low temperature. Hot deformation can destroy the (BOR) between the α and β phases and cause α and β phases to produce substructures. After a larger deformation, more substructures are generated, which will provide a higher driving force for subsequent globularization [41]. Deformation is also the necessary prerequisite for the globularization of lamellar α. Static globularization is a relatively slow process with the extending of annealing time, especially at the middle or late stage of static globularization [36]. Thus, compared to the other two factors, the sensitivity level of time is the smallest one.

The developed ANN model was applied to predict the static globularization kinetic curves at four prestrain (0.32, 0.53, 0.83, 1.06) and four annealing temperatures (750 °C, 775 °C, 800 °C, 825 °C). The static globularization behavior is a result of multifactor coupling, and GA/BP-ANN has excellent prediction capability for this kinetic process. As shown in Figure 14, the kinetic curves from 10 to 480 min were predicted by the developed ANN. The kinetic curves have an excellent fitting with the experimental data points, and the curves are smooth and stable without the overfitting phenomenon. The slope of curves increases with increasing annealing temperature and prestrain in the early annealing stage (before 60 min). When the annealing time exceeds 60 min, there is an obvious decrease in the globularization rate. As the microstructure evolution analyzed in the previous section (Section 3.2.2), the dominant mechanisms are termination migration and Ostwald ripening, which is a slower diffusion of atoms, causing the decrease in the globularization rate. High temperatures accelerate the migration of solution atoms, which is beneficial to the termination migration and the Ostwald ripening. Thus, the globularization rate after 60 min is improved when the annealing temperatures are 800 or 825 °C. It is worth mentioning that with the increase in annealing temperature, the difference value of the globularization fraction between the prestrain of 0.32 and other prestrains (0.53, 0.83, 1.06) gradually increases. According to the analysis in previous sections, the stored energy at the prestrain of 0.32 is lower than at the higher prestrain. When the annealing temperature increases, the sufficient release of stored energy leads to the improvement of the globularization fraction. At the prestrain of 0.32, the improvement of the globularization fraction is limited on account of lower stored energy. The lack of stored energy is the main reason for this phenomenon in the globularization fraction.

## 4. Conclusions

The wedge-shaped sheets of Ti-55511 alloy with lamellar microstructure were hot-rolled at 750 °C, and high-throughput rolled sheets were obtained. The samples with gradient strain distributions were annealed at 750, 775, 800, and 825 °C at different times. Then, the observation specimens were taken in the regions corresponding to true strains of 0.32, 0.57, 0.83, and 1.06 on the high-throughput annealed samples.

Static globularization is promoted with increasing prestrain, annealing temperature, and annealing time. The chain-like α phase formed when annealed at low temperatures. High annealing temperature boosts the globularization of lamellar α.The deformed lamellar α will furtherly generate new sub-boundaries. Boundary splitting occurs at LAGBs and HAGBs of lamellar α, and it can be accelerated when the annealing temperature increases.The GA/BP-ANN model can fit and predict the static globularization fraction well. The values of R^2^ for the training set and validation date are 0.99414 and 0.99111, respectively, and the AARE values are 3.17% and 3.22%, respectively. The trained GA/BP-ANN model has high prediction accuracy in predicting the static globularization kinetics during annealing without the occurrence of overfitting.The sensitivity level of influential factors was calculated using the trained GA/BP-ANN model. The order of sensitivity level for the globularization fraction is annealing temperature > prestrain > annealing time.

## Figures and Tables

**Figure 1 materials-16-01031-f001:**
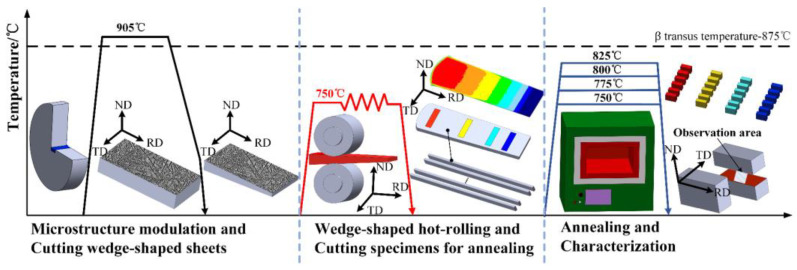
Schematic illustration of the experimental process.

**Figure 2 materials-16-01031-f002:**
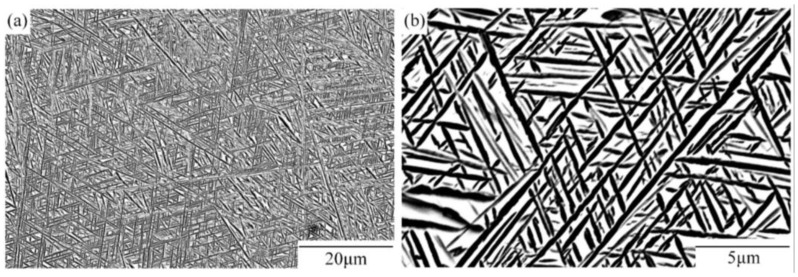
Microstructure of Ti-55511 alloy before hot rolling: (**a**) low and (**b**) high magnification.

**Figure 3 materials-16-01031-f003:**
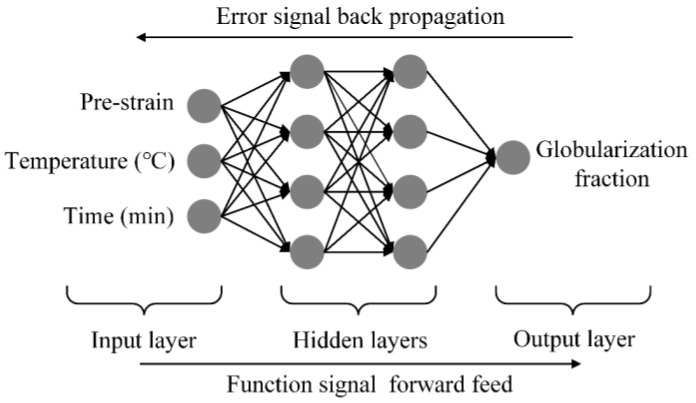
Schematic diagram of the BP-ANN model of static globularization.

**Figure 4 materials-16-01031-f004:**
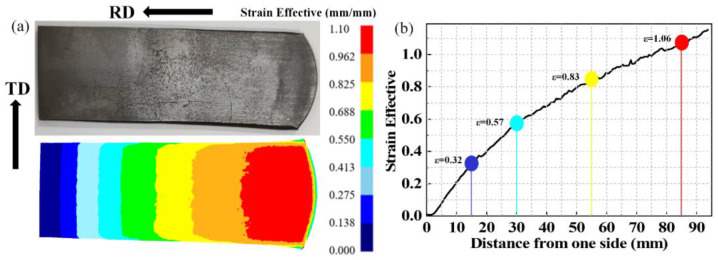
(**a**) The wedge-shaped hot-rolled sheet and its corresponding strain distribution on the mid-thickness plane; (**b**) specific strain value in observation areas.

**Figure 5 materials-16-01031-f005:**
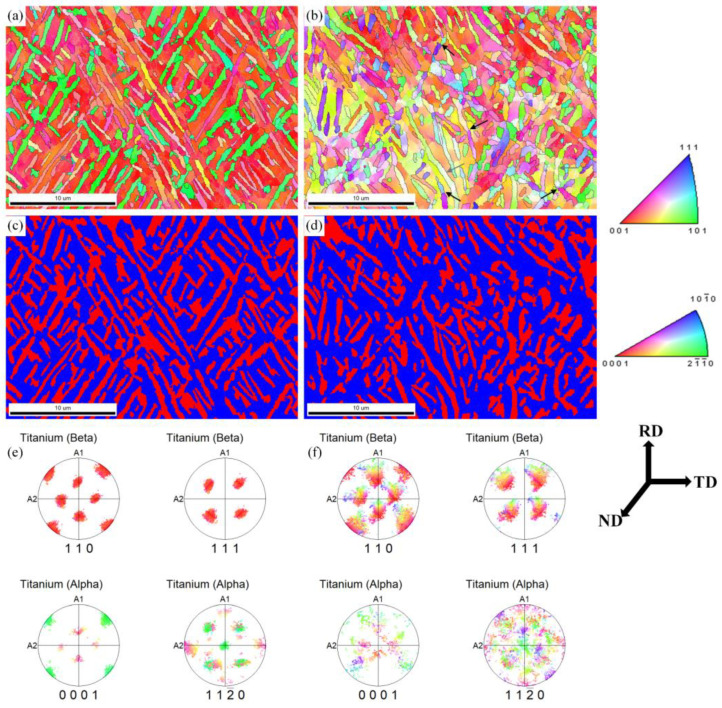
IPF color maps and phase maps of α and β phases with a strain of (**a**,**c**) 0.32, (**b**,**d**) 0.83; pole figure of α and β phase with a strain of (**e**) 0.32, (**f**) 0.83.

**Figure 6 materials-16-01031-f006:**
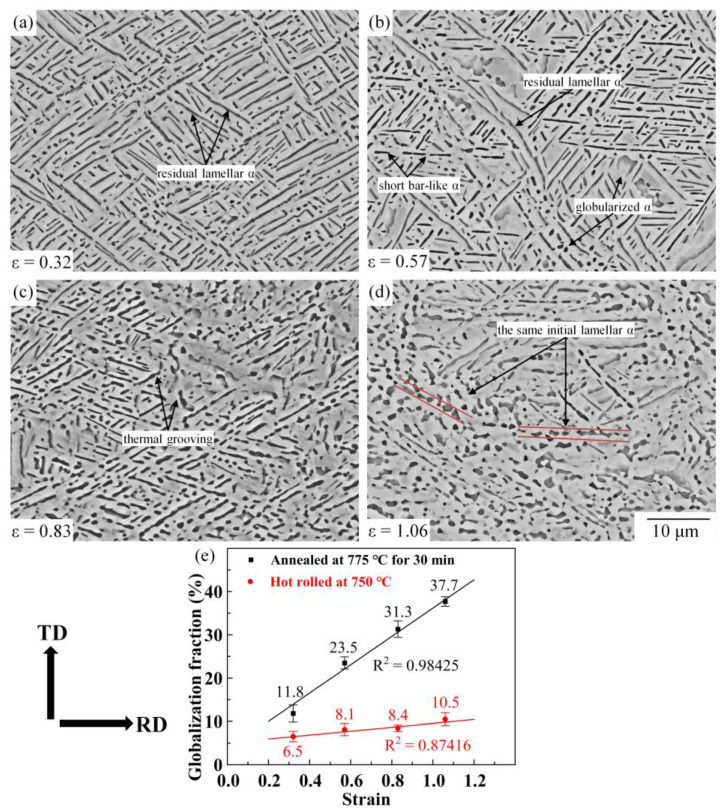
Microstructure of Ti-55511 alloy deformed to different strains and annealed at 775 °C for 30 min: (**a**) 0.32, (**b**) 0.57, (**c**) 0.83, (**d**) 1.06; and (**e**) the globularization fraction of α phase.

**Figure 7 materials-16-01031-f007:**
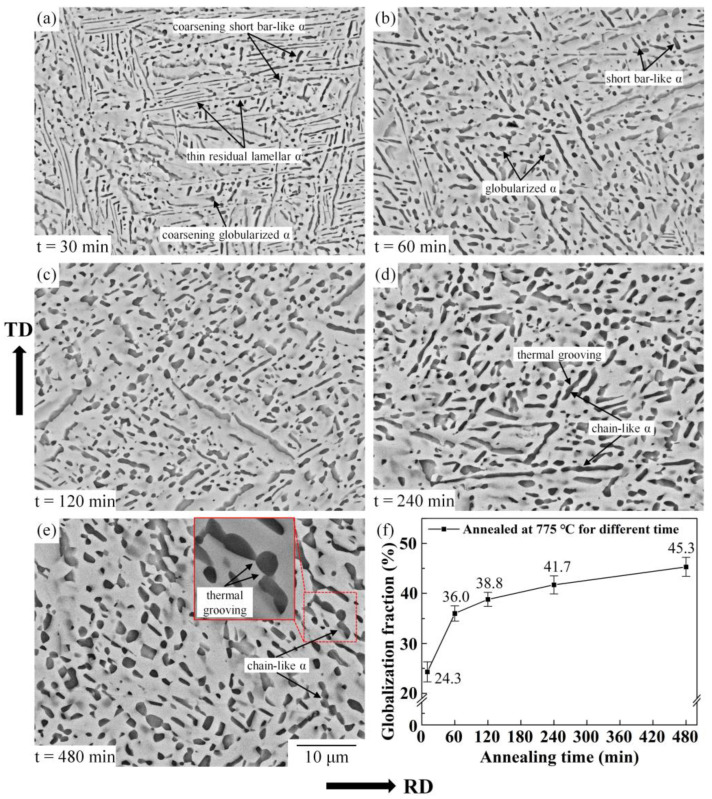
Microstructure of Ti-55511 alloy deformed to a strain of 0.83 and annealed at 775 °C for different times: (**a**) 10 min, (**b**) 60 min, (**c**) 120 min, (**d**) 240 min, (**e**) 480 min; and (**f**) the globularization fraction of α phase.

**Figure 8 materials-16-01031-f008:**
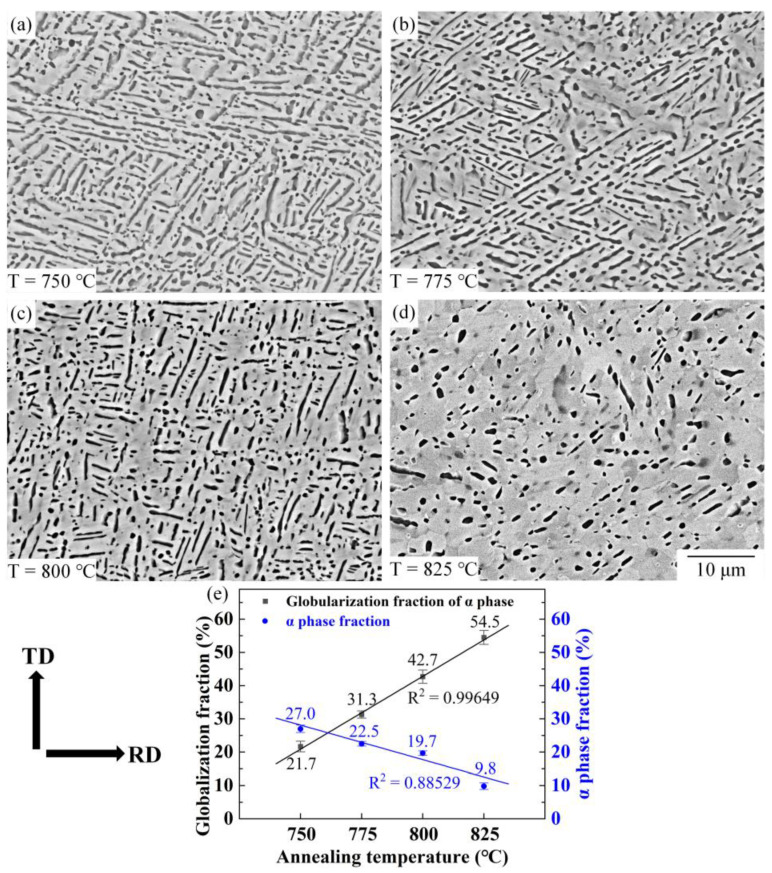
Microstructure of Ti-55511 alloy deformed to a strain of 0.83 and annealed at different annealing temperatures for 30 min: (**a**) 750 °C, (**b**) 775 °C, (**c**) 800 °C, (**d**) 825 °C; (**e**) the fraction and globularization fraction of α phase.

**Figure 9 materials-16-01031-f009:**
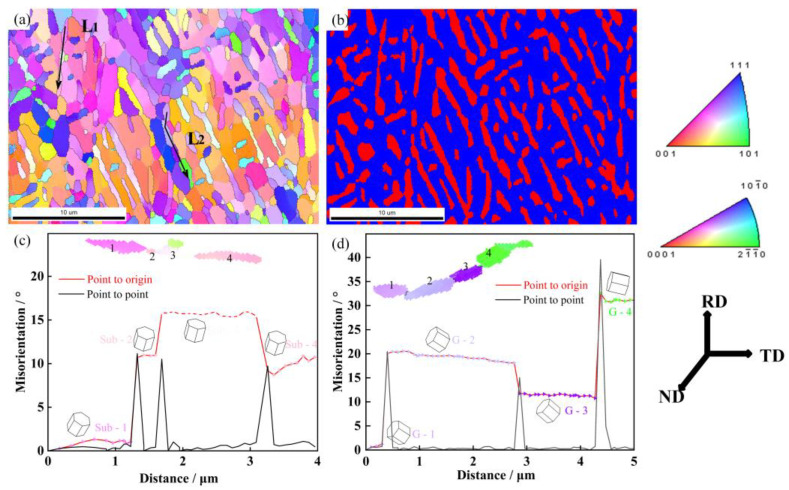
EBSD result of Ti-55511 alloy deformed to a strain of 0.83 and annealed for 30 min at 775 °C: (**a**) IPF map, (**b**) phase map. Misorientation change along (**c**) L_1_; (**d**) L_2_.

**Figure 10 materials-16-01031-f010:**
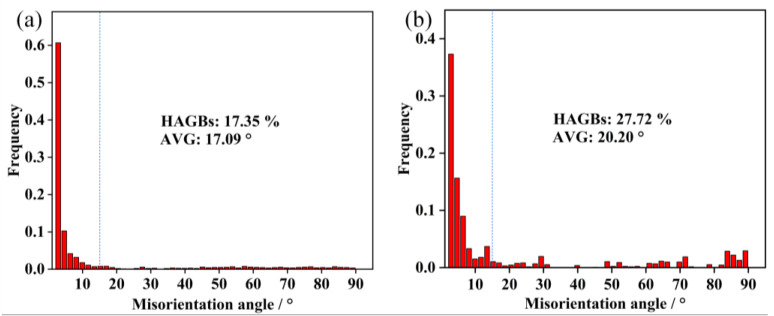
(**a**) Distribution of grain boundary misorientation for Ti-55511 alloy deformed to a strain of 0.83 at 750 °C and (**b**) annealed at 775 °C for 30 min.

**Figure 11 materials-16-01031-f011:**
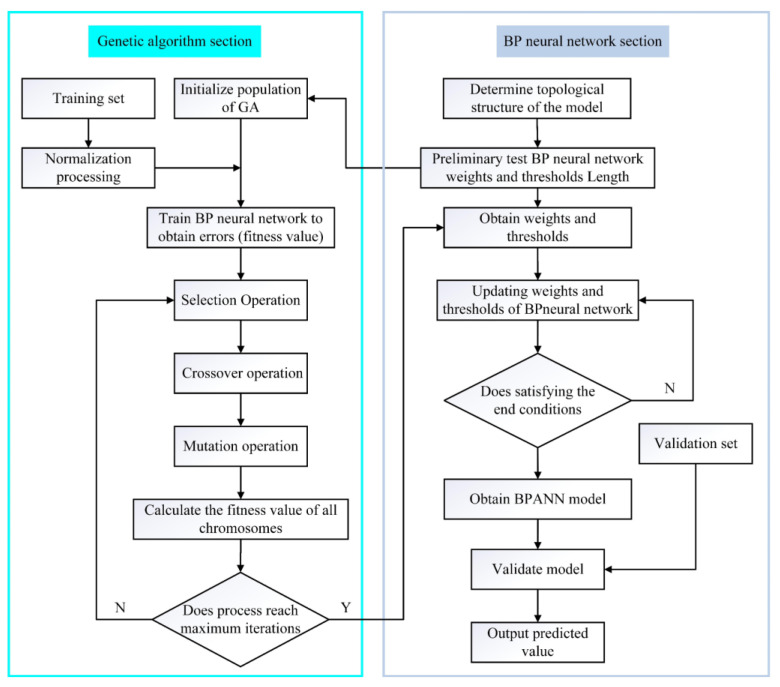
Flowchart of GA/BP-ANN.

**Figure 12 materials-16-01031-f012:**
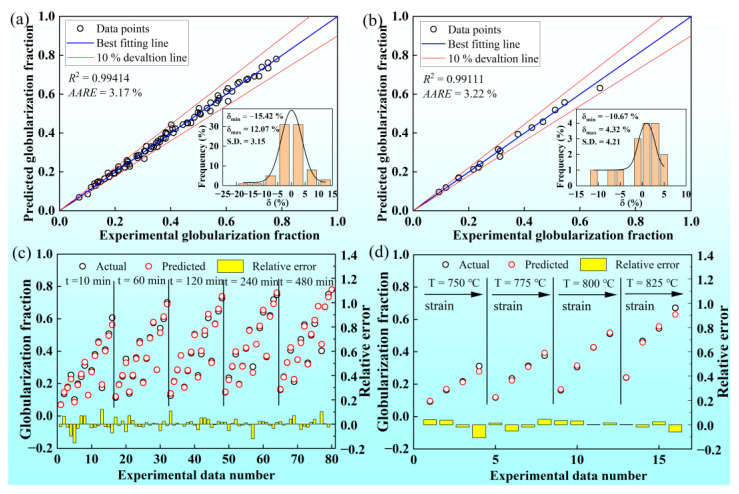
(**a**) Correlation and error analysis of the training set; (**b**) correlation and error analysis of the validation set; the relative errors of (**c**) the training set and (**d**) the validation set.

**Figure 13 materials-16-01031-f013:**
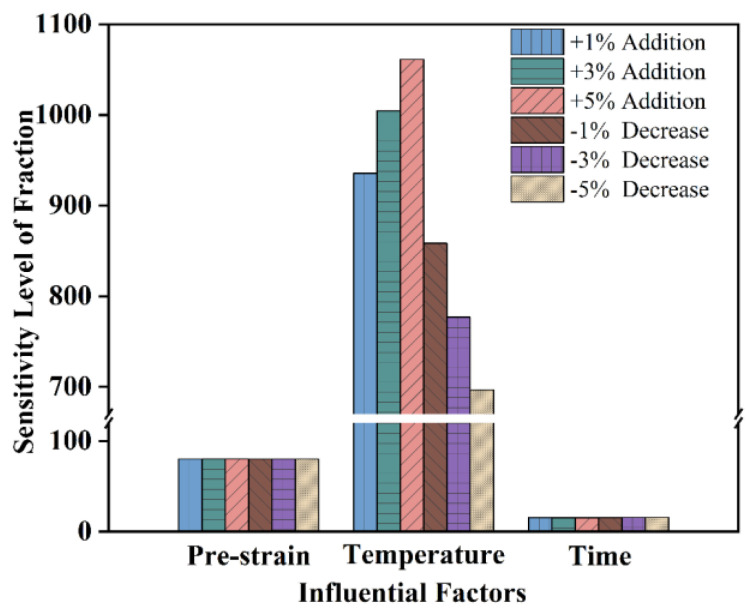
Sensitivity of input parameters to static globularization fraction.

**Figure 14 materials-16-01031-f014:**
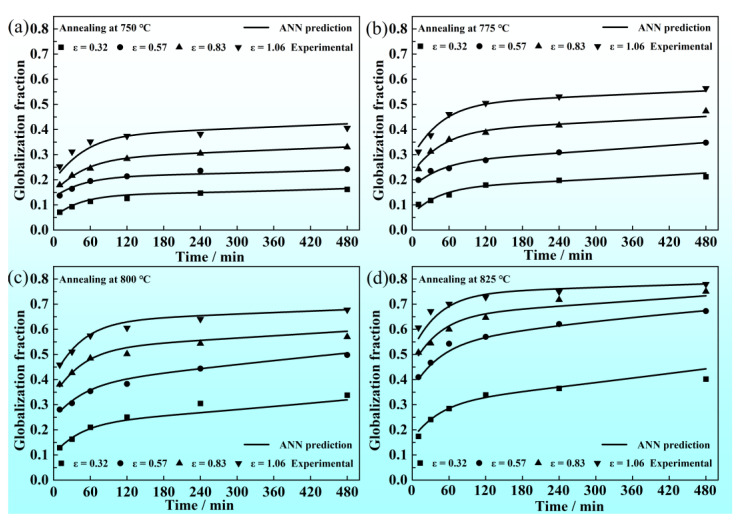
Static globularization kinetic curves predicted by GA/BP-ANN at different temperatures: (**a**) 750 °C; (**b**) 775 °C; (**c**) 800 °C; (**d**) 825 °C.

## Data Availability

All raw data supporting the conclusion of this paper are provided by the authors.

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
