# Peer review of "Static Globularization Behavior and Artificial Neural Network Modeling during Post-Annealing of Wedge-Shaped Hot-Rolled Ti-55511 Alloy"

_materials, 2023, doi:10.3390/ma16031031_

Round 1

Reviewer 1 Report

The authors explore a wide range of strain by the clever use of a tapered section subjected to rolling, which enables them to draw up maps of good combinations of properties for their Ti alloy. 

This reviewer is not competent to assess the analytical aspects of the submission, but aspects of the presentation are unnecessarily poor and are recommended to be re-drawn.  Figures 6, 7 and 8 are presented as bar charts.  Why do this?  Bar charts should, in general, never be used in scientific reporting.  A proper scientific graph is just as easily drawn and is far more informative, enabling accurate interpolation and extrapolation. An example is given of Fig 6 showing that the data when drawn as a graph reveal the annealing result to extrapolate accurately thro the origin, whereas the hot rolling data does not. Are these significant zeroth data information? (The example demonstration graph attached was roughly drawn on reused paper showing typing on the back of the sheet. Apologies for that.)

(In contrast, figures 9 and 12 are properly drawn scientific graphs.)

Incidentally, 'simi' (line 40) should read 'semi'.

Reviewer 2 Report

The study makes an essential contribution to the microstructural transformation of titanium in terms of globularization. It is thought that the academic and scientific aspects of the study are quite good. In addition, the experimental work and scientific approaches are quite original. Therefore, it is recommended that the study can be published after some minor revisions.

There are spelling and grammatical errors in the study; The study must be checked and corrected by a native speaker.

The representation of the first reference is incorrect. At the end of the first sentence of the introduction, marked "0".

When giving chemical composition in the material section, “wt.%” notation is appropriate, not “in wt.%”.

In the material section, "schematic illustration" is sufficient instead of "specific schematic illustration". Therefore, the specific representation is unnecessary.

The abbreviation of BOR should be given in the form of “Burgers orientation relationship” in the first place in the text.

Is any protective atmosphere used at 905 C during the heat treatment? If not, what is the effect in terms of impurity? In addition, the effect of chemical composition on globularization has not been sufficiently discussed.

Figure 4b is not clear. It is recommended to give the values on the curve in black.
